

# Boundary time crystals as AC sensors:
# Enhancements and constraints

Dominic Gribben[1], Anna Sanpera[2,3], Rosario Fazio[4,5],
Jamir Marino[1] and Fernando Iemini[1,6⋆]

**1** Institute for Physics, Johannes Gutenberg University Mainz, D-55099 Mainz, Germany
**2** Física Teòrica: Informació i Fenòmens Quàntics,
Universitat Autònoma de Barcelona, 08193 Bellaterra, Spain
**3** ICREA, Pg. Lluís Companys 23, 08010 Barcelona, Spain
**4** The Abdus Salam International Center for Theoretical Physics,
Strada Costiera 11, 34151 Trieste, Italy
**5** Dipartimento di Fisica "E. Pancini", Università di Napoli "Federico II",
Monte S. Angelo, I-80126 Napoli, Italy
**6** Instituto de Física, Universidade Federal Fluminense,
24210-346 Niterói, Brazil

⋆ fernandoiemini@id.uff.br

## Abstract

We investigate the use of a boundary time crystals (BTCs) as quantum sensors of AC fields. Boundary time crystals are non-equilibrium phases of matter in contact to an environment, for which a macroscopic fraction of the many-body system breaks the time translation symmetry. We find an enhanced sensitivity of the BTC when its spins are resonant with the applied AC field, as quantified by the quantum Fisher information (QFI). The QFI dynamics in this regime is shown to be captured by a relatively simple Ansatz consisting of an initial power-law growth and late-time exponential decay. We study the scaling of the Ansatz parameters with resources (encoding time and number of spins) and identify a moderate quantum enhancement in the sensor performance through comparison with classical QFI bounds. Investigating the precise source of this performance, we find that despite of its long coherence time and multipartite correlations (advantageous properties for quantum metrology), the entropic cost of the BTC (which grows indefinitely in the thermodynamic limit) hinders an optimal decoding of the AC field information. This result has implications for future candidates of quantum sensors in open system and we hope it will encourage future study into the role of entropy in quantum metrology.

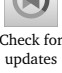

# 1  Introduction

Time crystals are non-equilibrium states which arise when quantum many-body interactions lead to the spontaneous breaking of a temporal symmetry present in the Hamiltonian. Such a symmetry breaking was originally proposed by Wilzcek [1] and the field has since flourished [2–5], beginning with an initial spate of theoretical studies on Floquet time crystals (FTCs) [6–11], so named as they exhibit a spontaneous ordering in time at a frequency subharmonic to a Floquet driving, which were soon followed by multiple experimental realizations [12–17].

The inclusion of dissipation opened an avenue of research into a class of time crystals breaking a *continuous* time symmetry [18]. The first case of these theorized, and the focus of this paper, is that of boundary time crystals (BTCs) [19]. Here the system lies on the boundary of a much larger environment and in the thermodynamic limit forms a macroscopic fraction of the total system plus environment constituents. Taking the environment to be Markovian (i.e., stationary) allows the system dynamics to be captured by time-independent Liouvillian; this dynamics exhibits a time-crystalline order. This order arises in the thermodynamic limit where the real part of the Liouvillian gap vanishes while the imaginary part survives resulting in persistent oscillations; there is no explicit frequency at which the system is driven.

Time crystals display robust coherence and correlations making them appealing candidates for application in various quantum technologies. In particular there have been several recent publications investigating their use in quantum metrology [20–24]. Generally, metrology seeks to optimally estimate a parameter through observing the statistics of a system (the sensor) affected by this parameter [25–28]. This parameter could be, for example, some intrinsic system property or the strength of an external field incident on the sensor. For sensing AC external fields in particular time crystals have been seen to be effective candidates. In Ref. [20] they showed that the persistent oscillations of an FTC allowed significant quantum enhancement in the sensing of an AC magnetic field incident on the system.

There is no direct connection that suggests that a BTC sensor has equal performance to an FTC. Despite having a few similar properties, such as long-lived oscillations with a decay rate that vanishes in the thermodynamic limit, they occur in rather different physical contexts and classes of time symmetry breaking. Moreover, unlike the FTC, the frequency of the long-lived oscillations in a BTC aren't contingent on any explicit external driving of the system but rather arise solely from the intrinsic energy scales present in the Liouvillian; this fact even increases the appeal of using the BTC. To further this enthusiasm there are also clear results showing a continual growth of genuine multipartite correlations as the BTC evolves [29,30], the presence of these make the potential for quantum enhancement much more feasible. However, as we shall show, the story is not so simple.

We study the performance of the BTC by computing the quantum Fisher information (QFI) [31] of the system in response to changes in the strength of the applied AC field; a larger QFI implies an improved sensor performance. Here we consider solely the detection of weak signals; specifically, we work in the linear response limit where the strength of the signal to measured is vanishingly small. There are various bounds on the QFI depending on factors such as the class of parameter being sensed [32,33], the structure of the Liouvillian [34–36], and correlations, or lack thereof, within the system [37]. These bounds are expressed in terms of the resources being utilised during the sensing protocol, in our case the relevant resources are the number of spins $N$ and the encoding time $t$. Given an open system sensing a time-dependent field, we rely on quite general bounds contingent on simply the presence of coherence and/or many-body correlations. For a classical system, with neither of these, the QFI is bounded by $Nt$; allowing for coherence increases this bound to $Nt^2$; and including many-body correlations gives us the Heisenberg limit: $N^2t^2$.

However, given that the bounds derived in presence of dissipation [34–36,38] are generally more restrictive, it should not be surprising if achieving the Heisenberg limit is impossible. The intuition behind this being that the entanglement and coherence necessary for quantum enhancement are famously fragile to dissipation; relatively simple examples can be constructed to show how introducing dissipation completely erases any quantum advantage from entanglement [39]. The need of precise metrological bounds in the presence of dissipation is of paramount importance, requiring deep numerical analysis such as the one presented here.

By computing the QFI of our protocol we find that its dynamics can be well-captured by a relatively simple Ansatz consisting of competing power-law growth and an exponential decay. In identifying this Ansatz we are able to better isolate the effects of any collective quantum enhancement (as would be evident in the initial growth) from the inevitable loss of information to the environment (as captured by the exponential decay). The timescale associated with this decay is found to scale with $1/N$, identical to the lifetime of the BTC itself, ensuring thus the capability of the sensor in arbitrary long single-shot protocols, in stark contrast with the short lifetime associated with typical open quantum systems. On the one hand, we observe that in the case of no constraints over the sensing time of the protocol,i.e., allowing the entire sensing period to be utilized, the maximum QFI scales with $N^2$. On the other hand, if the sensing time is fixed, the QFI scales superlinearly with the number of spins $N$, for large enough system sizes.

In comparison to known bounds, these results show that the BTC sensor displays a moderate quantum enhancement when detecting AC fields, exceeding the classical limit but, however, lying far from the Heisenberg limit. This implies that there is some hitherto unconsidered property of the BTC suppressing its sensing capability; we identify this property as the entropy. We observe that the entropy grows over the BTC lifetime, which diverges in the thermodynamic limit. These results seem at odds with a mean-field (MF) picture, usually employed for such classes of infinite-range interacting models. We recall, however, that the BTC sensor cannot be captured by a MF approach since it relies on many-body properties at the density matrix level, rather than simply the (one-body) magnetization.

The paper is structured as follows. In Section 2 we first introduce the model we consider, namely the known BTC Liouvillian with a time-dependent AC field term, before introducing the key quantity of interest of this paper, and quantum metrology in general, the quantum Fisher information (QFI). In Section 2.1 we discuss the resonant properties and the set of field parameters we will consider in our analysis of the sensor performance. The most relevant parts of our studies are contained in Section 3, where we explore the dynamics of the QFI and derive a proper Ansatz for it, and in Section 4 where we provide the explanation of the observed behaviour linked to the dynamical properties of the BTC. We present our main conclusion in Sec. 5.

## 2 Model

We consider a system of $N$ spin-1/2 particles that are evolving under a collective drive and dissipation. The dynamics of this system is described by the permutation-symmetric master equation:

$$\partial_t \hat{\rho}(t) = \hat{\mathcal{L}}_B[\hat{\rho}(t)] = -i\omega_0[\hat{S}^x, \hat{\rho}(t)] + \frac{\kappa}{(N/2)}\left(\hat{S}^-\hat{\rho}(t)\hat{S}^+ - \frac{1}{2}\{\hat{S}^+\hat{S}^-, \hat{\rho}(t)\}\right), \qquad (1)$$

where the $N$ spins are driven coherently with strength $\omega_0$ and decay collectively with a rate of $\kappa$ rescaled by $N/2$. We have introduced collective spin operators $\hat{S}^{\alpha=x,y,z} = \sum_i \hat{\sigma}_i^\alpha/2$ where $\hat{\sigma}_i^\alpha$ is the Pauli-$\alpha$ matrix acting on the $i$-th spin. These also describe a spin of length $S$ with raising/lowering operators $\hat{S}^\pm = \hat{S}^x \pm i\hat{S}^y$

In the thermodynamic ($N \to \infty$) limit, this model undergoes a dynamical phase transition as a function of $\omega_0/\kappa$: when $\omega_0/\kappa < 1$ the spins decay to a steady state but for $\omega_0/\kappa > 1$ they oscillate in perpetuity. This non-trivial phase corresponds to a breaking of the continuous time-translation symmetry of Eq. (1), occurring only in the thermodynamic limit and, in this way supporting a time crystal phase. Specifically this is known as a Boundary Time Crystal (BTC), or alternatively as a dissipative time crystal, referring to the fact that the spins are only a subsystem of the total system-environment Hilbert space. Despite being a portion of the whole system, the boundary spins can be macroscopically large, and stabilise different phases of matter. Importantly, although the total system-environment Hamiltonian is time-translation invariant —consistent with various different derivations of the above effective master equation (see, e.g., supplementary information of Ref. [19] or Refs. [40–42]) — the boundary spins can still break this symmetry, leading to persistent oscillatory behaviour and thus supporting a time crystal phase.

The goal of this work is to utilise the long-lived oscillations of the BTC to enhance the sensing of an AC field. This is inspired by Ref. [20] where evidence of quantum-enhanced sensing was seen in a FTC.

The full model of the BTC in the presence of AC field is given by

$$\partial_t \hat{\rho}_g(t) = \hat{\mathcal{L}}_B[\hat{\rho}_g(t)] - ig[\hat{H}_{ac}(t), \hat{\rho}_g(t)], \qquad (2)$$

where the AC Hamiltonian is

$$\hat{H}_{ac}(t) = \hat{S}^z \sin(\omega_{ac}t + \phi). \qquad (3)$$

Here, the AC field is characterized by its frequency, $\omega_{ac}$, and its phase shift, $\phi$. We have also introduced $\hat{\rho}_g(t)$: the density matrix conditioned on the value of the field strength, $g$, evolved to time $t$. Our observations indicate that choosing the external field along the z-direction leads to higher values of sensor performance as compared to the x- or y-direction, making it a more effective choice for the sensing protocol. Although we have no formal analytical proof that this direction is always optimal, the key insight stems from the instability of the time crystal phase - or its higher susceptibility - to perturbations along the z-direction. This instability has been demonstrated in analytical studies within a mean-field treatment [43] and observed in generalised versions of the model [44,45]. As a consequence, the system exhibits a stronger response to the applied AC field, thus allowing a greater precision in its estimation.

To determine the sensitivity of the BTC to variations in the strength of this field $g$ we compute the quantum Fisher information (QFI). The QFI bounds the uncertainty in estimating the value of $g$, with $\Delta g \geq 1/\sqrt{MF(\hat{\rho}_g(t))}$, where $F(\hat{\rho}_g(t)$ is the QFI of the sensor probe, and $M$ is the number of measurements performed in the system in order to extract and therefore

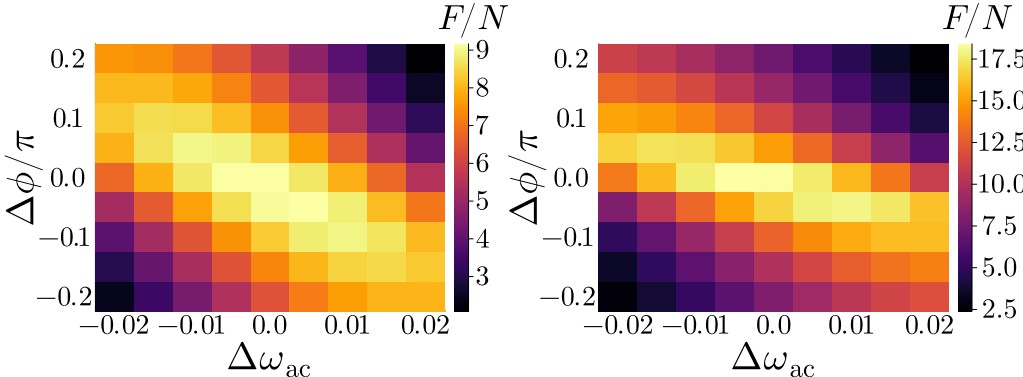

Figure 1: The maximum QFI achieved during a single dynamics for various detunings of both the phase, $\phi$, and AC field frequency, $\omega_{\mathrm{ac}}$, with $N = 128$ (left) and $N = 256$ (right). For both of these plots we set $\omega_0 = 4\kappa$.

decode the information about $g$. Clearly, a larger QFI magnitude indicates an improved sensor performance. This is given by [46, 47]

$$F_g(\hat{\rho}) = 2 \sum_{k,l} \frac{|\langle \lambda_k | \partial_g \hat{\rho}_g | \lambda_l \rangle|^2}{\lambda_k + \lambda_l}, \tag{4}$$

where we have introduced the density matrix $\hat{\rho}_g$, depending on a real parameter $g$ and $\{\lambda_i\}$ are the eigenvalues of $\hat{\rho}_g$. Here we have suppressed the time dependence for simplicity.

Computing $\partial_g \hat{\rho}_g$ can be done numerically by either a finite difference in terms of $g$ or, as we shall show, by computing a second master equation in conjunction with Eq. (2). We can find the time evolution of $\partial_g \hat{\rho}_g$ by simply taking the time derivative of Eq. (2), this given by

$$\partial_t \partial_g \hat{\rho}_g(t) = \hat{\mathcal{L}}_B[\partial_g \hat{\rho}_g(t)] - i[\hat{H}_{\mathrm{ac}}(t), \hat{\rho}_g(t) + g \partial_g \hat{\rho}_g(t)]. \tag{5}$$

In this equation, notice that $\partial_g \hat{\rho}_g(t)$ is dependent on $\hat{\rho}_g(t)$, via the second term, but the converse is not true. In the linear response limit, it is therefore possible to use this master equation to compute $\partial_g \hat{\rho}_g(t)$ for any $\hat{H}_{\mathrm{ac}}(t)$ with a single solution of $\hat{\rho}_{g \to 0}(t)$. However, although we remain in the linear response regime for all results presented here, we found it more convenient to solve both Eqs. (2) and (5) simultaneously, as it is simpler to allow an adaptive solver to pick the best timestep for the global problem as we vary $\hat{H}_{\mathrm{ac}}(t)$.

Some properties of the QFI are worth recalling. While for classical systems the QFI is limited simply by number of repetitions in the estimation protocol, leading to a linear scaling in time (one can think of partitioning the whole time into small time bins representing the number of measurements), exploiting quantum coherence in the system allows a quadratic improvement over time [48], extending the maximun of QFI to $F_g(\hat{\rho}) < t^2$. Furthermore, since Fisher Information is additive, a sensor composed of $N$ separable spins obeys the bound $F_g(\hat{\rho}) < N t^2$. Surpassing this bound requires nonseparability, that is, harnessing quantum correlations among the spins within the sensor. The ultimate quantum advantage emerges when entanglement is fully exploited, allowing a quadratic gain in both the number of spins and time, thus achieving the Heisenberg limit with $F_g(\hat{\rho}) = N^2 t^2$.

## 2.1 Resonant fields

To sharpen the focus of this paper, we restrict the study to a particular set of AC field parameters: those that result in optimal sensing in the $N \to \infty$ limit. We anticipate that the response of the system at large $N$ is maximal when the AC field is resonant to the internal dynamics

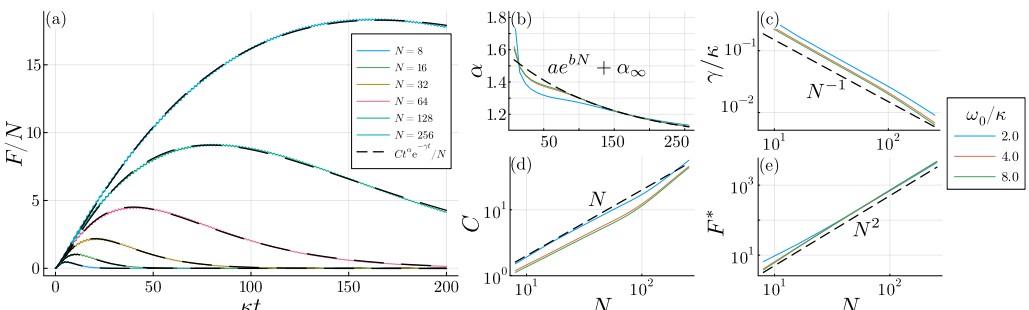

Figure 2: Analysis of the sensor performance via the QFI dynamics. (a) Dynamics of the rescaled QFI for various $N$ at a fixed $\omega_0 = 4\kappa$, the dashed lines correspond to a fit of the Ansatz $C t^\alpha e^{-\gamma t}$. The values of these fit parameters as a function of $N$ are plotted in (b)-(d) for various $\omega_0$ with dashed lines to help indicate the trends. (e) Maximum of the QFI over the entire evolution as a function of $N$ with a dashed line again to indicate the scaling.

of the BTC spins. In the thermodynamic limit ($N \to \infty$) this corresponds to a frequency $\omega_{\mathrm{ac}} = \omega_{\mathrm{BTC}} \equiv \sqrt{\omega_0^2 - \kappa^2}$ and phase $\phi = \phi_{\mathrm{BTC}} \equiv \sin^{-1}(\kappa/\omega)$ [30]. In Figure 1 we plot, for $N = 128$ and $N = 256$, the maximum of the QFI during its evolution, as a function of shifts around these resonant values, i.e., $\omega_{\mathrm{ac}} = \omega_{\mathrm{BTC}} + \Delta\omega_{\mathrm{ac}}$ and $\phi = \phi_{\mathrm{BTC}} + \pi\Delta\phi$. We see that on increasing $N$ the overall maximum QFI approaches the point $\Delta\phi = \Delta\omega_{\mathrm{ac}} = 0$, apart from finite size corrections of order $O(1/N)$. Therefore, although these values will not be optimal for all $N$, for simplicity we fix $\omega_{\mathrm{ac}}$ and $\phi$ to their thermodynamic limit resonant values and investigate the performance of the sensor as a function of the resources $N$ and $t$.

## 3 QFI dynamics

In this section, we study the scaling of the QFI as a function of $N$ and $t$. We use its scaling behaviour to characterize the performance of the sensor and compare this performance with the limits of a classical sensor as well as the fundamental "Heisenberg" quantum limit. These limits are expressed in terms of the total resources used in the sensing protocol. Incorrect characterization of the resources used or omitting a certain resource, such as sensing time, can lead to results which seemingly surpass even the Heisenberg limit. We shall be careful to state which resources we consider in each result. All results presented here correspond to an initial state for which $\langle \hat{S}^z \rangle = N/2$, they were generated using QuantumOptics.jl [49] and code adapted from this framework.

The dynamics of the QFI (scaled by $N$) for various $N$ is plotted in Figure 2(a). These dynamics follow the typical behaviour of the QFI for a Markovian open quantum system [39, 50]: it initially increases over time before reaching some peak value and decaying. This initial growth arises from the coherence in the direction of the applied field accruing a phase. The coherence, however, only survives for a finite time due to the system being open; as it decays, so does the ability to encode information. Consequently the QFI ceases growth, reaching a peak and subsequently decaying; we find that the time this peak occurs is related to the lifetime of the underlying time crystal dynamics.

We find that the envelope of these dynamics can be captured by a relatively simple functional form given by

$$F(t) = C t^\alpha e^{-\gamma t} . \tag{6}$$

This fit, again rescaled by $N$, is also shown in Figure 2(a). This Ansatz was inspired by previous analytical results on quantum metrology in open quantum systems [39]. The parameters of this Ansatz completely characterize the QFI dynamics. We study the scaling of the parameters $\alpha$, $\gamma$ and $C$ to better determine the performance of the sensor as a function of the resources $N$ and $t$.

First, in Figure 2(b), we study the scaling of $\alpha$ as a function of $N$ for various $\omega_0$ and obtain that the initial growth exponent decreases as the system size is increased; we are able to capture the large $N$ behaviour with an exponential decay to some asymptotic value that we label $\alpha_\infty = \lim_{N\to\infty} \alpha$. So, while the overall performance of the sensor is clearly superior for larger systems, as evidenced by Figure 2(a), the initial growth exponent is reduced. This is of significance as the exponent with which the QFI grows has been shown to be linked to the degree of enhancement due to quantum correlations; a classical system is limited to grow linearly in time. The classical bound arises because over a single sensing run, the only resource available is the sensing time. There is little dependence on the driving, $\omega_0$, except in the region of small $N$ and $\omega_0$. We attribute this effect to the shifting of the peak to shorter times, combined with a decrease of oscillations' frequency while their amplitude increases. As such in this region we are attempting to fit a power-law to short times where the oscillations, rather than the envelope, will dominate the value of $\alpha$ we extract. We also find a dependence of $\alpha$ on $\kappa$, independent of $\omega_0$, which we discuss in Section 4.

The decay rate of the QFI, $\gamma$, scales like $1/N$ independent of $\omega_0$, as can be seen in Figure 2(c). This coincides with the rate at which the dissipative gap closes in the time crystal phase [19] indicating that the ability to encode information on the time crystal depends on its lifetime. This implies that increasing the size of the system not only improves its sensitivity but also its sensing time, thus allowing for an arbitrarily long single shot run. This is in sharp contrast with typical open systems sensors which can only work for a relatively short time.

In terms of the Ansatz parameters, the time taken for the QFI to reach its maximum during the evolution is given by

$$t^* = \frac{\alpha}{\gamma}. \tag{7}$$

Given that $\alpha \sim a\exp(-bN) + \alpha_\infty$ and $\gamma \sim N^{-1}$ we have that $t^* \sim N$ for large $N$. This scaling is also of relevance to the performance of the sensor in terms of $\alpha$, in that while it is true that decreasing system size improves the growth exponent at short times, the region in which this growth occurs shrinks like $1/N$.

The amplitude of the QFI, $C$, is plotted in Figure 2(d) and scales linearly with $N$ at small $N$ but appears to become superlinear as $N$ is increased. From the scaling behaviour of $\alpha$ and $\gamma$, we see that at large $N$ both $t^\alpha$ and $e^{-\gamma t}$ become independent of $N$. Therefore, in this limit the scaling of $C$ corresponds to the scaling of $F$ at a fixed sensing time, i.e., our only resource is $N$, indicating quantum enhancement.

We now study how the peak of the $F$, labelled $F^*$, scales with $N$. In doing so we are disregarding time as a constraint in the sensing protocol; this is shown in Figure 2(d). Largely we find that $F^*$ scales quadratically with $N$ and at large $N$ there is a suggestion that the scaling even exceeds this. We recall that while the height of the peak may scale greater than $N^2$, the time to reach the peak, $t^*$, scales with $N$. And so in terms of total resources the Heisenberg limit is not breached. This result is, nevertheless, of relevance to experiments where sensing time can be freely varied.

Overall we see that while our BTC sensor exhibits a quantum-enhanced performance, it is still far from the optimal Heisenberg limit. In the remainder of this paper we aim to understand the physical mechanisms behind the BTC sensor performance. In particular, what are the sources of its improvements? And what are the sources of its deficiencies?

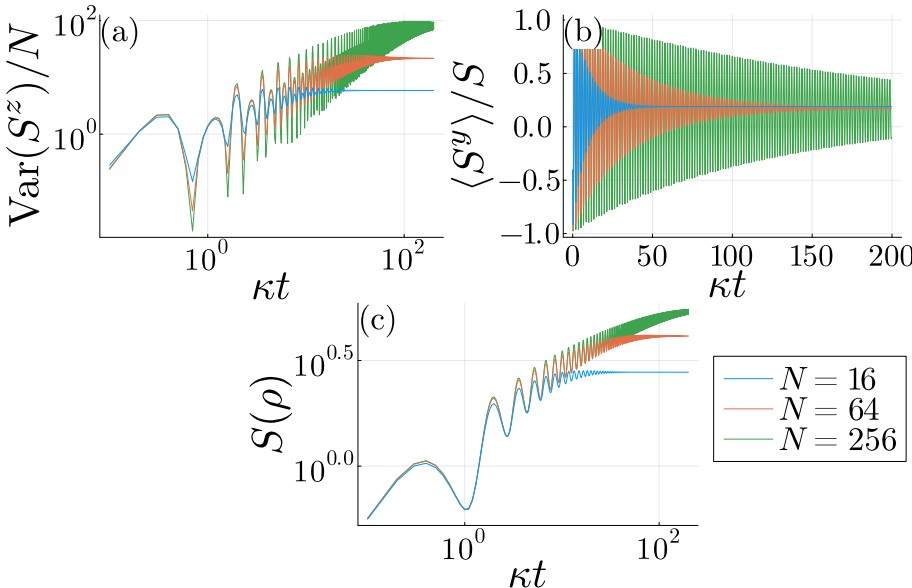

Figure 3: Dynamics of different BTC properties associated with its sensor performance. (a) The rescaled variance of the magnetization driven by the AC-field, (b) the coherence present in the $z$-basis, i.e. the direction of the applied field, and (c) the dynamics of the von Neumann entropy. We set $\omega_0 = 4\kappa$ for all these results.

# 4 Understanding the QFI scaling

We consider two quantities whose presence we expect to signify an enhanced sensitivity of the BTC sensor to an applied AC field, namely the variance and coherence of the system in the direction of this field. These can be directly related to the QFI for the case of time-independent sensing with pure states. Pure states are known to be the optimal choice for probes [51], but at finite $N$ we don't expect our state to remain pure. With this in mind we also compute the entropy dynamics to detect the degree to which the BTC departs from this optimum. Finally, we consider the behaviour of the exponent $\alpha$ in the thermodynamic limit to make a connection with the non-interacting and closed system limit.

We first attempt to understand the QFI dynamics by studying the correlations within the system, specifically in the form of the variance of the total spin operator in the direction of the applied AC field. This quantity corresponds e.g., to the QFI of a closed system exposed to a DC field acting in the $z$-direction [31]. We plot the variance, $\text{var}(\hat{S}^z) = \langle(\hat{S}^z)^2\rangle - \langle\hat{S}^z\rangle^2$, rescaled by $N$ in Figure 3(a). We also consider the coherence present in the eigenbasis of the applied field. While a large variance generally corresponds to the ability to encode a greater degree of phase information, the phase in the basis can only be affected if there is some coherence. We show in Figure 3(b) the coherence in the $z$-basis, as captured by $\langle\hat{S}^y\rangle$. We observe that both the variance growth rate and coherence amplitude at a given time increase with $N$, beneficial properties for metrology and therefore related to sources of improvements for the BTC sensor. Moreover, the lifetime of these quantities scale with $N$, making a direct correspondence to the QFI decay rate $\gamma$. Despite such advantageous properties, these results seem to contradict the decreasing scaling we observe of $\alpha$ with $N$. This contradiction can be resolved by considering the relevant role of the entropy.

As we can see from Figure 3(c), the entropy grows at a rate which approaches some constant, finite value in the thermodynamic ($N \to \infty$) limit, showing that the quantum state is intrinsically mixed during its whole evolution. In other words, there is an entropic cost towards the stabilization of the BTC phase. Given the concavity of the QFI [51], we can expect

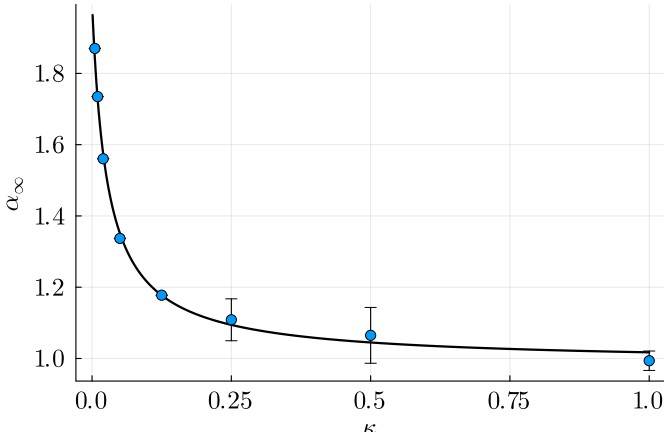

Figure 4: We show the thermodynamic limit ($N \to \infty$) value obtained for the exponent $\alpha$, for varying decay rate, $\kappa$. This exponent tends to 2 in the closed system limit ($\kappa \to 0$) and it is independent of $\omega_0$. The black line is there to guide the eye, we were unable to find a good Ansatz for this dependence.

this increase in entropy to hinder the encoding of information on the system. This argument is supported by considering how the growth exponent in the thermodynamic limit, $\alpha_\infty$, varies on reducing the dissipation rate $\kappa$, i.e., as we approach a closed system with entropy conservation. We observe in Fig. 4 that upon reducing the dissipation and its detrimental entropic costs for the QFI, $\alpha_\infty$ indeed becomes larger. In the limit $\kappa \to 0$ the exponent approaches the value of 2, i.e. the bound for closed non-interacting systems.

*Failure of mean-field (MF) approach.–* An $\alpha = 2$ exponent is also what one would expect – for any value of $\kappa$ – by considering the MF approach in order to describe the system dynamics. In this approach the spins are considered non-interacting and, moreover, with a conserved entropy [45]. We recall that, although a MF approach is usually employed for such classes of infinite-range interacting systems, its validity is restricted to which observables one aims to describe. For example, while for magnetization observables (one-body observables) it gives exact results in the thermodynamic limit [52–54], more complex observables at the level of the density matrix cannot be captured by MF – such as the QFI. It is currently unclear, however, how to distinguish this class of observables from those for which the MF approach fails. We explore further these issues by computing the QFI of the MF density matrix and show explicitly how the exact result diverges from the MF limit over time.

We first recall that the MF equations of motion for the BTC in the presence of an AC field in the $z$-direction are obtained closing the second order cumulants, $\langle \hat{S}^\alpha \hat{S}^\beta \rangle = \langle \hat{S}^\alpha \rangle \langle \hat{S}^\beta \rangle$, and are given by

$$\partial_t \langle \hat{S}^x \rangle = \frac{\kappa}{S} \langle \hat{S}^x \rangle \langle \hat{S}^z \rangle - g \langle \hat{S}^y \rangle \sin(\omega_{\text{AC}} t + \phi),$$

$$\partial_t \langle \hat{S}^y \rangle = -\omega_0 \langle \hat{S}^z \rangle + \frac{\kappa}{S} \langle \hat{S}^y \rangle \langle \hat{S}^z \rangle + g \langle \hat{S}^x \rangle \sin(\omega_{\text{AC}} t + \phi), \tag{8}$$

$$\partial_t \langle \hat{S}^z \rangle = \omega_0 \langle \hat{S}^y \rangle - \frac{\kappa}{S} (\langle \hat{S}^x \rangle^2 + \langle \hat{S}^y \rangle^2),$$

where we neglect terms of order smaller than O(N). The total density matrix in this limit can be described by the factorized Ansatz as

$$\hat{\rho}_{\text{MF},g} = \bigotimes_{i=1}^{N} \hat{\rho}_{i,g}, \tag{9}$$

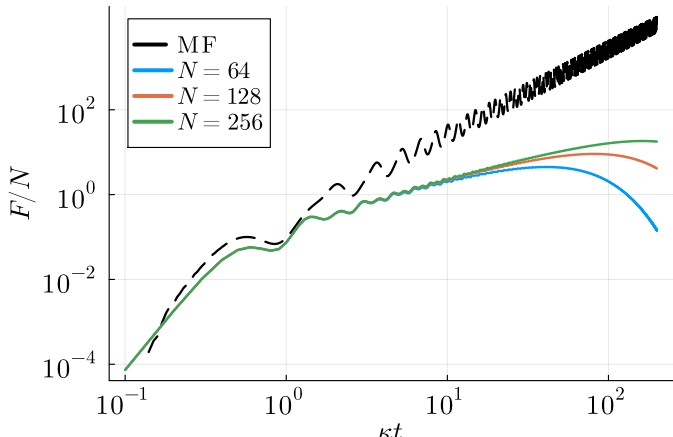

Figure 5: Comparison of exact QFI dynamics for finite system sizes $N$ with the one obtained by the MF approach in the thermodynamic limit ($N \to \infty$), both in linear response regime ($g \to 0$). The AC-field is set in resonance to the BTC and $\omega_0 = 4\kappa$. The MF results are computed from Eq.(12) with displacement $dg = 0.01$.

where $\hat{\rho}_{i,g}$ is the reduced density matrix for the $i$-th spin given the AC field $g$. These reduced states are equivalent for all spins and can be expressed in terms of the total spin observables,

$$\hat{\rho}_{i,g} = \mathbb{I}_2/2 + (\langle \hat{S}^x \rangle \sigma^x + \langle \hat{S}^y \rangle \sigma^y + \langle \hat{S}^z \rangle \sigma^z)/N \,. \tag{10}$$

In this way we compute the total QFI by using the identity for separable systems, $F(\hat{\rho}_A \otimes \hat{\rho}_B) = F(\hat{\rho}_A) + F(\hat{\rho}_B)$, which implies

$$F(\hat{\rho}_{\mathrm{MF},g}) = N F(\hat{\rho}_{i,g}), \tag{11}$$

and determine numerically the QFI of the reduced state in terms of the fidelity, namely,

$$F(\hat{\rho}_{i,g}) = \lim_{dg \to 0} 8 \left( 1 - \mathrm{Tr}\left[ \sqrt{\sqrt{\hat{\rho}_{i,g+dg}} \hat{\rho}_{i,g} \sqrt{\hat{\rho}_{i,g+dg}}} \right] \right) \,. \tag{12}$$

In Figure 5 we compare this quantity in the linear response limit to the the exact QFI computed for finite $N$, with the the AC-field parameters set to those resonant with the BTC. Up to times $t \sim \kappa^{-1}$ both approaches yield similar results with small quantitative differences. Beyond this point up to times $t \sim N^{-1}$ the exact QFI converge to a result distinct from the MF limit. The MF result grows indefinitely like $F \sim t^2$ as is expected for coherent oscillations. These results further substantiate the claim of the main text that the sensor performance is suppressed by entropy growth as these approaches only begin to diverge on timescales on which entropy begins to significantly grow.

## 5 Conclusions

We have studied the performance of a BTC functioning as a sensor of an external AC magnetic field by computing the QFI dynamics. We were able to identify a simple Ansatz which captured the essential features of these dynamics. This Ansatz consisted of three parameters: a power-law growth exponent $\alpha$, an exponential decay rate $\gamma$, and an overall scaling constant $C$. We found that $\gamma$ and $C$ follow intuitive trends: $\gamma$ vanishes like $1/N$, while $C$ grows linearly at small $N$ and superlinearly at large $N$. In comparison to known bounds, these results suggest that the BTC sensor shows a moderate quantum enhancement over an arbitrarily long timescale,

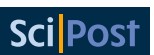

exceeding the classical limit, however lying far from the Heisenberg limit. Furthermore, $\alpha$ was shown to decrease with increasing system size and asymptote to a value less than that predicted by the MF description expected to hold in the thermodynamic limit. To investigate the origin of this performance we considered the dynamics of BTC properties relevant to metrology.

While a BTC supports long time coherence and variance correlations, leading to collective enhancement, the sensor performance is hindered by the entropic cost of its dynamics. This picture is evidenced by the entropy dynamics shown in Figure 3(c); and further substantiated in Fig. 4 where by decreasing the dissipative rate $\kappa$, corresponding to a decrease in entropy growth, we achieve an improved growth exponent for the QFI.

We also compare the exact QFI dynamics to the usually employed MF approach for such classes of inifite-range systems, showing explicitly their non equivalence. While a MF Ansatz predicts coherently oscillating spins, with a quadratic QFI growth ($\sim t^2$) the exact dynamics features a slower and quantitatively different dynamics. These results further substantiate the claim of sensor performance suppression due to entropy production.

In future it would be interesting to investigate the performance of the BTC as an AC sensor in the vicinity of its critical point; there are many examples of enhanced sensitivity near criticality [23, 55–59] and evidence that the associated scaling is robust to periodic driving [60]. Beyond this, we hope that our work encourages a deeper study into the role of entropy in quantum metrology.

# Acknowledgments

The authors gratefully acknowledge the computing time granted on the supercomputer MOGON 2 at Johannes Gutenberg-University Mainz (hpc.uni-mainz.de).

**Data availability:**  The data and codes of this manuscript are available from the corresponding author upon reasonable request.

**Funding information**  F.I. acknowledges financial support from Alexander von Humboldt foundation and the Brazilian funding agencies CAPES, CNPQ, and FAPERJ (Grants No. 308205/2019-7, No. E-26/211.318/2019, No. 151064/2022-9, and No. E-26/201.365/ 2022) and by the Serrapilheira Institute (Grant No. Serra 2211-42166). A.S. acknowledges financial support from the Spanish Agencia Estatal de Investigacion, Grant No. PID2022-141283NB-I00, the European Commission QuantERA grant ExTRaQT (Spanish MICINN project PCI2022-132965), by Ministerio de Ciencia e Innovación with funding from European Union NextGenerationEU(PRTR-C17.I1) and by Generalitat de Catalunya and the Spanish Ministry of Economic Affairs and Digital Transformation through the QUANTUM ENIA project call – Quantum Spain project, and the European Union through the Recovery, Transformation and Resilience Plan – NextGenerationEU within the framework of the Digital Spain 2025 Agend. R.F. ackowledges financial support from PNRR MUR project PE0000023-NQSTI and by the European Union (ERC, RAVE, 101053159). D.G. and J.M. acknowledge financial support from the Deutsche Forschungsgemeinschaft (DFG, German Research Foundation) via TRR 306 QuCoLiMa ("Quantum Cooperativity of Light and Matter"), Project-ID 429529648 (project D04) and in part by the QuantERA II Programme that has received funding from the European Union's Horizon 2020 research and innovation programme under Grant Agreement No 101017733 ("QuSiED"). Views and opinions expressed are however those of the author(s) only and do not necessarily reflect those of the European Union or the European Research Council. Neither the European Union nor the granting authority can be held responsible for them.

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
