# Peer review of "Boundary Time Crystals as AC sensors: enhancements and constraints"

_SciPost Physics, doi:SciPost Phys. 18, 100 (2025)_

## Round 2 · Referee Report · Anonymous (Referee 1) · 2024-11-27

Report
The authors investigate the application of boundary time crystals for the purpose of AC field quantum sensing. In this model, a number of two level systems (spins) evolve according to a specific, permutation-symmetric master equation with a collective drive and dissipation. This system has a non-trivial phase in the thermodynamic limit, depending on the ratio of the drive and decay parameters. If the driving is strong enough, permanent oscillations occur asymptotically. In a recent paper by some of the current authors, it has been demonstrated that Floquet time crystals may be employed for the same purpose. In analogy to the case of Floquet time crystals, the quantum Fisher information measure (QFI) is used as a tool to assess the possibility of enhanced sensing.
A numerical analysis of the QFI behavior (considering the resonant case and optimal phase corresponding to the thermodynamic limit) reveals that it is possible to fit a power-exponential curve as a function of time, for various system sizes N and frequencies. The authors the examine various BTC properties and identify the role of entropy in this context. Namely, the entropy grows and reaches some finite value in the thermodynamic limit, thus the system is not in a pure quantum state (entropic cost). The entropic cost may be decreased by decreasing dissipation. The authors also compare the results of a mean field calculation, providing some insight how on larger time-scales the exact results deviates from the MF calculations, although on small time-scales they both coincide.
The presented calculations lead to the conclusion that BTC sensors possibly provide a moderate quantum enhancement, however it cannot be exculded that around the critical point of the system, higher performance can be achieved. I wonder, why and how much the sensitivity could be enhanced around the critical point, since a brief explanation would add a lot to the analysis of the paper. This is just an optional question. It is possible that the answer is complicated and could be addressed in another publication.
In conclusion, I find the manuscript very well written, interesting, about a timely and important topic, thus I suggests its publication in SciPost Physics.
Recommendation
Publish (easily meets expectations and criteria for this Journal; among top 50%)
Author: Dominic Gribben on 2025-02-18 [id 5233]
(in reply to Report 2 on 2024-11-27)
Dear Referee,
We thank you for taking the time to read our manuscript and for your kind feedback. Regarding the behaviour of the sensor near critical point, it is quite a technically challenging question to address. As the model is brought closer to the critical point a larger system size is needed such that the dynamics persist long enough to see oscillations. The period of these oscillations also increases meaning a longer simulation time would be needed to encode the external field strength. Both of these facts lead to greater computational demands. Thank you for raising this, as we mention in our conclusion we agree that it is an interesting question but believe it to be beyond the scope of this paper.
Yours Sincerely,
Dominic Gribben - on behalf of the authors
Author: Dominic Gribben on 2025-02-18 [id 5232]
(in reply to Report 1 on 2024-11-28)Dear reviewer, thank you for taking the time to read and provide feedback on our manuscript. Please find our more detailed response in attachment.
Attachment:
sensing_BTC_refresponse.pdf

---

## Round 2 · Referee Report · Anonymous (Referee 2) · 2024-11-28

Strengths
The manuscript addresses an innovative and relevant topic, proposing the use of boundary time crystals (BTCs) for metrological purposes, specifically as quantum sensors for AC fields. This novel application of BTCs is likely to inspire further investigations in quantum metrology, as indicated by the manuscript’s existing 7 citations, which underscores its resonance with specialists in the field. Additionally, identifying entropy production as a limitation in achieving the Heisenberg limit adds valuable insights into the interplay of dissipation and metrological performance.
Weaknesses
The primary weakness of the manuscript is its lack of self-contained explanations in certain sections, which could hinder accessibility for a broader audience. Expanding some discussions and providing additional references would enhance the work's clarity and comprehensibility. Furthermore, inconsistencies in normalization conventions across figures and a limited discussion of certain assumptions weaken the overall presentation.
Report
The manuscript is well-structured, presenting a comprehensive numerical investigation of the boundary time crystal (BTC) model and exploring its potential as a sensor for AC fields. It demonstrates the feasibility of BTCs for quantum-enhanced sensing, leveraging their long-lived coherence and multipartite correlations. However, the results indicate that while BTC sensors surpass classical limits, they fall significantly short of achieving the Heisenberg limit, with entropy production identified as the primary factor limiting performance. By connecting these findings to the broader context of quantum metrology, the manuscript offers valuable insights into the role of dissipation and coherence in open quantum systems.
I recommend a minor revision to address these concerns. After these revisions are done I believe the present work will meet the journal's acceptance criteria.
Requested changes
1. Model Explanation:
For readers unfamiliar with time crystals, it is unclear why this model constitutes a time crystal. The statement "Specifically this is known..." following Equation (1) is vague. I suggest expanding this section with additional references or providing a brief derivation in an appendix to substantiate the claim.
2. Discussion of Quantum Fisher Information (QFI):
Extending the discussion of QFI in Equation (4) to include the expected scales and their implications for the sensor’s performance would be welcome. For instance, explaining the significance of F/N > 1 as a quantum enhancement and why F/N \approx N indicates the Heisenberg limit. Although some of this discussion is already present in the introduction, I believe that moving and expanding it would benefit the coherence of the manuscript. I also think that the introduction of the operator $\hat{L}_g$ seemed unnecessary since it was not used in practice to obtain the results.
3. Normalization in Figures:
I would recommend using consistent normalization of the figures. For example, in Figure 1 F was normalized to N^2, while Figures 2 and 5 normalize F to N. Alternatively the authors could justify, in each figure, their choice for the given normalization.
4. Choice of External Field Orientation:
The external field is assumed to be oriented in the z-direction, as defined in Equation (3). Clarify whether this choice is standard or made for convenience. If it is standard, cite appropriate literature. If not, discuss the implications of considering arbitrary driving directions.
5. Entropy Suppression:
Does the parameter \omega_0 (coherent driving strength) influence entropy production? Could modifying the driving Hamiltonian, e.g., introducing Heisenberg interactions, suppress entropy production and improve performance?
6. Code Availability:
To ensure reproducibility of the results I would suggest, not require, the authors to publish also simulation codes on a public repository such as GitHub. This aligns with the transparency encouraged by the SciPost community and will benefit future researchers working on related topics.
7. Symbol Usage:
The same symbol `S` is used for total spin length and entropy. This could confuse readers. Consider using a distinct symbol for the spin operator.
8. Typo:
On page 6. "... they were were generated ... " has too many "were" -s.
Recommendation
Ask for minor revision

---

## Round 3 · Author Response

Please find attached the revised manuscript titled ``\textit{Boundary Time Crystals as AC sensors: enhancements and constraints}'' We are grateful to the reviewers for their time and insightful comments. We have carefully considered all the feedback and detailed our responses below. We hope with these changes, the article is now ready for publication.
Yours Sincerely,
Dr. Dominic Gribben - on behalf of the Authors

---

## Round 3 · List of Changes

-
Expanded the discussion following Equation (1) in order to clarify the nature and terminology of boundary time crystals. Additional references have also been included to provide a broader theoretical context. See paragraph ``Specifically this is known as... supporting a time crystal phase.''.
-
To improve coherence, we have expanded the discussion in Section 3 regarding the expected scaling of the QFI and its implications. See paragraphs
The QFI bounds the uncertainty in estimating the value of $g$... decode the information about $g$'' andA few bounds of the QFI ... Heisenberg limit with $F=N^2 t^2$.'' Additionally, we agree that the operator $\hat L_g$ was not used directly in obtaining our results, and therefore we have removed it. -
We have ensured consistency in figure normalizations. Now, all figures with QFI data use a uniform normalization of $F/N$ for clarity and comparability.
-
We have now included a discussion on our choice of z-direction for the external field, along with references to related studies. See paragraph ``Our observations indicate that...a greater precision in its estimation.''
-
We have included the following statement in the manuscript: ``The data and codes of this manuscript are available from the corresponding author upon reasonable request.''
-
We have removed the usage of S as total spin length, adding explictly its value when needed.

---

## Editorial Decision

published